# Assessing the impact of human error factors on railway accident severity: Evidence from accident investigation reports in Korea

Changhun Kim, Jun Lee*

Division for Road Transport Research, Korea Transport Institute, Sejong, South Korea

* junlee@koti.re.kr

## Abstract

This study investigates the role of human error factors in shaping the severity of railway accidents. Using a structured coding scheme to transform qualitative accident investigation reports into quantitative variables, the analysis reveals that deficiencies in managerial oversight, shortcomings in maintenance practices, and failures in equipment and system reliability are consistently associated with higher accident costs. These findings underscore the organizational and technical dimensions of human error as critical factors linked to accident severity, rather than merely front line worker mistakes. At the modeling level, the explanatory variables collectively account for a meaningful portion of the variation in accident costs, indicating that such outcomes are not purely random but systematically related to underlying human and organizational conditions. By identifying the relative importance of managerial, maintenance, and equipment-related deficiencies, this study provides empirical evidence for prioritizing these factors in railway safety management. The results highlight the need to strengthen organizational accountability, improve maintenance regimes, and ensure the reliability of technical systems as essential strategies for mitigating high-consequence accidents. These insights contribute to a deeper understanding of the organizational and systemic correlates of accident severity and can inform targeted interventions aimed at enhancing overall railway safety.

## 1. Introduction

Railway systems are widely recognized as complex socio-technical infrastructures in which safety emerges from the interaction of technical components, human operators, and organizational structures [1]. Despite continuous advances in automation, digital signaling, and system engineering, human and organizational factors remain persistent sources of risk in railway operations [2]. According to the National Safety Council (NSC) [3], between 2015 and 2024, railway incidents in the United States resulted in an annual average of approximately 3,800–4,300 cases when fatal and

**Data availability statement:** All data and code files are available from the Zenodo repository at https://doi.org/10.5281/zenodo.18310486.

**Funding:** This research was supported by a Korean Agency for Infrastructure Technology Advancement (KAIA) grant funded by the Ministry of Land, Infrastructure and Transport (grant no. RS-2023-00239464)." We also confirm in the revised Funding Statement that the funder had no role in study design, data collection and analysis, decision to publish, or preparation of the manuscript.

**Competing interests:** The authors have declared that no competing interests exist.

non-fatal events among railway employees are combined (Fig 1). This sustained level of workforce casualties indicates that risks related to human and organizational factors remain substantial.

Railway employees—including operators, signal staff, maintenance workers, and construction personnel—are routinely exposed to high-risk environments [4–5], where even relatively minor human errors can escalate into severe incidents [6]. In addition to injuries and fatalities, such accidents are associated with service disruptions and economic losses [7], making workforce safety an important empirical issue in railway systems.

Recent studies have shown that railway accidents are rarely the result of a single technical failure and are more often associated with combinations of human error, organizational deficiencies, and management-related shortcomings [8–9]. Previous work has further emphasized the importance of human factors in safety-critical signaling operations [10]. In this context, official accident investigation reports are widely used as primary information sources because they document accident circumstances, causal chains, and contributory factors in detail.

Although research on railway safety and human factors has expanded, much of the existing literature remains qualitative or case-based. Only limited efforts have been made to transform narrative accident investigation reports into structured quantitative datasets suitable for large-sample statistical analysis. One example is a previous study that demonstrates the feasibility of extracting structured risk factors from railway accident reports [11]. Nevertheless, investigation reports continue to be used far more often for qualitative analysis and safety recommendations than for quantitative modeling.

Moreover, most existing studies focus on accident occurrence or accident causation mechanisms rather than on accident severity measured in economic terms, such as accident cost and material damage. Accident severity expressed in financial terms represents an important dimension of railway safety impact; yet, only a small number of studies have examined severity-related outcomes in the railway context. Among them, one study provides one of the few empirical analyses linking accident characteristics to financial losses [12]. Recent review work also indicates that large-sample, quantitative, report-based analyses focusing on accident severity remain relatively rare [13].

Against this background, the purpose of this study is not to validate or benchmark a classification scheme itself, but to use a structured coding scheme as an analytical tool to examine which types of human and organizational errors are more strongly associated with accident cost severity. This study focuses on workforce-related railway accidents and transforms qualitative findings from official accident investigation reports into a structured data set for statistical analysis. By doing so, it provides empirical evidence on the relative importance of different categories of human and organizational deficiencies in relation to accident severity and contributes to a more systematic, data-driven understanding of high-consequence railway accidents.

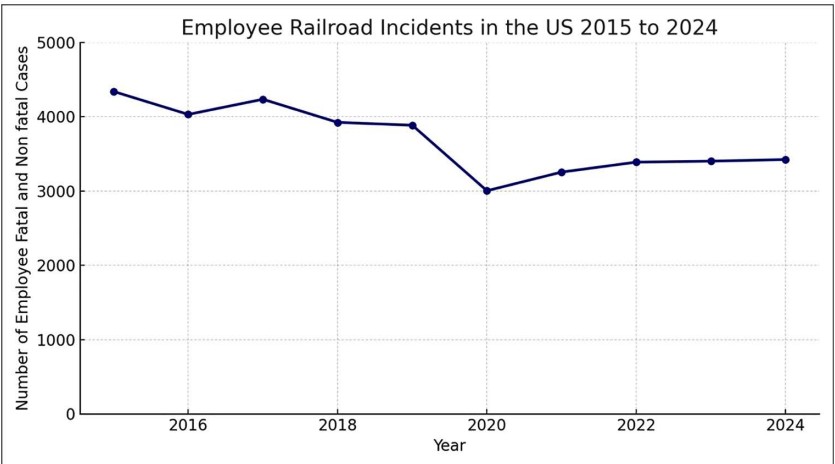

**Fig 1. Employee railroad incidents in the United States (2015–2024).** Source: NSC Injury Facts (2024).

## 2. Literature review

### 2.1 Human error factors among railway workers

Human error has long been recognized as a dominant cause of accidents across high-risk sectors, yet systematic research focusing specifically on railway employees remains limited. Early railway safety research tended to emphasize technical causes, including mechanical breakdowns or signaling failures [14–15], while human and organizational deficiencies were treated as secondary considerations.

However, empirical evidence from multiple national contexts underscores the central role of human and organizational deficiencies in railway safety. Analyses of U.S. incidents have repeatedly identified inadequate supervision as a contributing factor [16–17]. In Australia, human error has been confirmed as a causal factor in official investigation reports [18], while in South Africa, high accident rates have been linked to weak supervision and a negative safety culture [19]. Formal inquiries into major rail accidents have also highlighted communication breakdowns and hierarchical "authority gradients" among workers as causal contributors [20]. In addition, the U.S. Federal Railroad Administration has reported that accident and injury rates in railroad yards significantly exceed industry averages, prompting research into workforce safety issues such as safety climate and training [21]. Regulatory oversight itself has also been noted as an important influence on employee safety outcomes [22]. Collectively, these findings demonstrate that supervisory practices, organizational culture, and human factors remain fundamental drivers of accident risk across diverse railway systems.

A growing body of research has also explored the influence of safety culture and organizational attitudes toward risk. Comparative studies indicate that organizations with proactive reporting systems, continuous safety education, and participatory decision-making processes tend to have lower accident rates [23]. Conversely, rigid hierarchical communication structures and punitive approaches to error reporting may exacerbate risks by discouraging disclosure of near misses [24]. Despite these insights, the literature on railway workers is fragmented and often qualitative, offering descriptive accounts of accidents without robust quantification of the relative importance of different human error categories [25]. This lack of systematic frameworks for classification and measurement underscores the need for more rigorous, data-driven approaches to evaluating human error factors in railway operations.

## 2.2 Accident report data and analytical frameworks

Accident investigation reports are widely regarded as one of the most informative yet underutilized data sources for safety research [26–27]. They typically provide detailed accounts of accident circumstances, contributing factors, and outcomes, but the narrative and semi-structured nature of these documents often makes them difficult to analyze systematically. In other transportation sectors, researchers have addressed this limitation by transforming raw reports into structured datasets, which in turn have enabled large-scale statistical analysis [28]. For example, in aviation safety studies, narrative reports of crew errors have been systematically classified into error categories [29], while in road safety, police crash reports have been reorganized to capture human, environmental, and technical factors [30]. These efforts have revealed recurring causal patterns, such as driver distraction in road crashes and communication failures in aviation incidents [31].

In the railway sector, however, the use of accident reports has been more limited. Existing studies have often remained qualitative or descriptive, emphasizing case studies that highlight maintenance deficiencies, aging equipment, or procedural violations without translating them into variables suitable for quantitative evaluation [32]. A small number of studies have attempted to apply structured approaches, but these have rarely extended beyond basic categorization or summary statistics [33]. Compared with the more developed practices in aviation and road safety, railway research still lacks systematic frameworks for coding and analyzing the human error content embedded in accident reports [34].

This absence of consistent methods for structuring and analyzing the human error content of railway reports illustrates the current limitations of the field. While accident investigation documents provide rich qualitative descriptions, they have seldom been transformed into formats that allow systematic evaluation of workforce-related error factors.

## 2.3 Research gap and contributions

Despite growing recognition of the importance of human error in railway safety, existing research has remained largely descriptive and has not developed systematic frameworks for classifying or quantifying worker-related factors. At the same time, although accident investigation reports provide rich qualitative accounts of incidents, their potential as a data source for quantitative modeling has not been fully realized. These limitations have restricted the ability of prior studies to determine which categories of human error most strongly influence accident outcomes.

This study seeks to address these gaps in several ways. First, it shifts the focus of railway safety research toward the workforce, providing an evidence-based perspective on employee safety that has often been overlooked. Second, it consolidates diverse human error factors into a structured framework, enabling both systematic classification and empirical analysis. Third, by transforming narrative accident reports into a structured database and applying quantitative modeling techniques, the study demonstrates how the relative influence of different error categories on accident severity can be rigorously assessed. Finally, the findings provide a basis for advancing evidence-informed approaches to railway workforce safety. By identifying the relative influence of different error categories, the study supports the design of comprehensive strategies that may include organizational, technical, and human-centered measures, with potential applications in supervisory systems, maintenance planning, and workforce health management.

## 3. Data

### 3.1 Data source and case selection

The primary data set for this study consists of official railway accident investigation reports issued by the Aviation and Railway Accident Investigation Board (ARAIB) of the Republic of Korea [35]. These reports are mandated under the Act on the Investigation of Aviation and Railway Accidents, which establishes a standardized and independent investigation process oriented toward safety improvement rather than fault attribution. We collected and reviewed all railway accident investigation reports published between 2011 and 2025, thereby ensuring more than a decade of data produced under a consistent institutional and legal framework.

Under Article 4 of the Act, the Investigation Board is responsible for collecting and analyzing accident-related information, determining causes, and issuing safety recommendations. The scope of reportable accidents is defined in Article 2 and includes events such as collisions, derailments, operationally disruptive fires, accidents involving three or more casualties, and cases with property damage exceeding 50 million KRW. Importantly, an accident is subject to investigation if it satisfies any one of these criteria, rather than all of them simultaneously. Accordingly, the investigation reports cover a broad range of accidents defined by both the nature and the severity of the event under a unified legal framework.

Each report follows a standardized structure comprising a factual section and an analytical section. The factual section documents the event chronology, casualties and damages, personnel and operational information, and technical data on rolling stock, track, electrical, and signaling systems. The analytical section integrates these elements to evaluate human performance, infrastructure conditions, and organizational procedures. The final report concludes with an official determination of causes and safety recommendations. Fig 2 presents examples of accident evidence included in these reports, providing visual documentation that supports the assessment of both causal factors and damage severity.

The dependent variable in this study, accident cost, is constructed directly from the official damage assessment reported in Section 1.2 (Damage Information) of the factual section of each report, which consists of Section 1.2.1 (Casualties), Section 1.2.2 (Property Damage), and Section 1.2.3 (Other Damage). Because casualties are reported only in terms of the number of deaths and injuries and are not monetized, the cost measure used in this study is defined as the sum of property damage and other damage. Property damage includes losses related to track, rolling stock, signaling systems, and structures and facilities, while other damage covers compensation and economic losses associated with service disruptions, such as line closures and train delays. These amounts are officially determined and finalized by the Investigation Board through on-site investigations, technical testing, and deliberative procedures, and are published in the final investigation reports.

Because the same legal framework, reporting format, and assessment procedures have been applied consistently throughout the sample period, the accident cost measure provides a reasonable degree of temporal comparability across years, although it should be understood as an official ex post damage assessment rather than an accounting-based valuation.

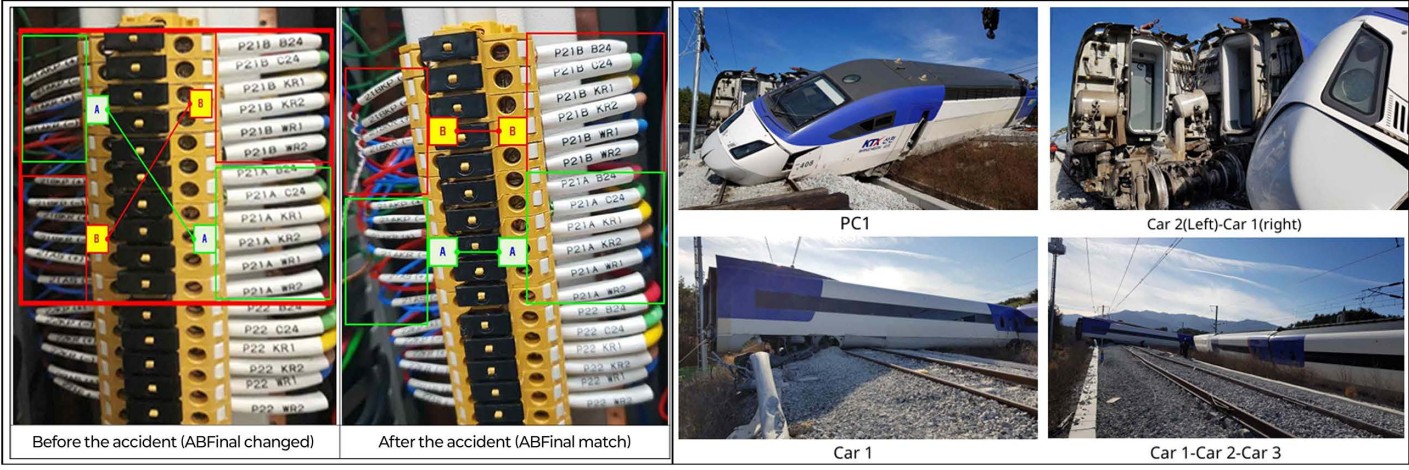

**Fig 2. Examples of accident evidence from investigation reports.** (Left) Signal equipment room terminal block before and after the accident. (Right) Condition of the accident train at the time of the incident. Source: Aviation and Railway Accident Investigation Board (ARAIB), Republic of Korea. Images reproduced for illustrative purposes.

With respect to sample construction, a total of 143 railway accident investigation reports were published by ARAIB during the study period. Among these, 105 cases were included in the final analytical sample, while 38 cases were excluded based on explicit exclusion criteria. First, cases in which no accident cost was reported or could be identified were excluded, such as incidents involving derailments without any physical damage or measurable property loss. Second, duplicate reports referring to the same accident (e.g., re-investigations or revised versions of previously published reports) were removed to avoid double counting. Third, unresolved cases in which the accident causes were not conclusively determined at the time of reporting were excluded. As a result, the final sample consists of 105 accidents for which both reliable cost information and sufficiently clear causal analyses were available for subsequent coding and quantitative analysis.

With respect to explanatory variables, each report specifies both the primary cause of the accident and a set of contributory factors. This standardized structure provides a consistent basis for identifying and coding human, technical, and organizational error factors, ensuring transparent and methodologically consistent mapping from qualitative investigation findings to predefined error categories.

## 3.2  Conceptual framework and coding scheme

To transform the qualitative content of accident investigation reports into analyzable data, this study adopts a structured classification framework for human and organizational error factors. Particular attention is directed to the sections describing official accident causes and contributory factors, as these elements are formally identified by investigators and therefore provide a consistent and authoritative basis for identifying human error variables.

Based on prior studies in railway safety and human factors, as well as a review of established human error classification frameworks across safety-critical domains, contributory factors are classified into eight categories (C1–C8). These categories represent recurring human and organizational deficiencies documented in accident investigations, including supervision and management, training and education, maintenance and inspection, equipment and emergency systems, safety regulation and compliance, communication, decision-making and competence, and physical and health conditions.

The conceptual foundations of each category and their correspondence with existing human reliability analysis frameworks are summarized in Table 1, which provides the theoretical rationale supporting the validity of the C1–C8 classification scheme. By grounding the classification in both prior literature and the institutional logic of accident investigation practice, this framework is intended to capture the major dimensions of human and organizational contributions to railway accidents in a systematic and reproducible manner.

## 3.3  Coding procedure and reliability

For empirical coding, each category was operationalized using observable indicators derived from investigation reports. Table 2 presents representative phrases and investigation findings that were used to assign a value of "1" to each category, thereby clarifying how abstract human factor concepts were translated into report-based coding decisions. These indicators were not treated as an exhaustive checklist but served as guiding criteria to ensure consistent interpretation across cases. When at least one indicator corresponding to a category was explicitly identified as a cause or contributory element in the report, the category was coded as present (1); otherwise, it was coded as absent (0).

To enhance coding reliability, the reports were independently reviewed and coded by two members of the research team in separate coding rounds. Inter-rater agreement was assessed using Cohen's kappa, which indicated a high level of consistency between coders (mean $\kappa = 0.88$, range: 0.62–0.98; see Table 3).

Discrepancies were subsequently examined through joint discussion, and final coding decisions were determined through consensus. In addition, an external railway safety expert reviewed the coding scheme and a subset of representative case assignments to validate the conceptual consistency of the coding logic. Cases that remained ambiguous after the initial comparison—particularly those involving borderline distinctions between training and competency deficiencies

**Table 1. Basis for defining categories (C1–C8) from prior human error classifications.**

| Human Error | HFACS (Aviation domain) | RSSB (Rail domain) | HPEP (Nuclear domain) |
|---|---|---|---|
| C1 | Unsafe Supervision | Competence Management (Interpretive) | Management Oversight |
| C2 | Organizational Influences (training elements) | Knowledge, Skills, Abilities emphasis (domain focus) | Implicit in Performance Evaluations |
| C3 | Preconditions (technical environment) | Infrastructure/System condition (Interpretive) | Task Environment (technical focus) |
| C4 | Preconditions (physical environment) | Task/Workplace system factors (domain focus) | Human-System Interface |
| C5 | Organizational Influences (process) | Procedures and guidance domain (Interpretive) | Implicit Regulatory/ Procedural Evaluation |
| C6 | Preconditions (group factors) | Communications (domain focus) | Communications/ Coordination & Control |
| C7 | Unsafe Acts (decision/policy level) | Decision and competency (Interpretive) | Decision-related measures in performance Evaluation |
| C8 | Preconditions (physiological) | Workload/Fatigue domain focus (domain focus) | Fitness/Health aspects in performance Evaluation |

**Note.** The coding categories (C1–C8) were developed based on and conceptually aligned with established human factors frameworks, including HFACS [36], the RSSB human factors guidance [37], and the HPEP framework [38–39].

(C2) and adjacent categories such as supervisory control (C1) or operational decision-making (C7)—were flagged and re-evaluated through joint discussion with expert consultation until final coding decisions were reached by consensus. The full coding scheme with category definitions and illustrative examples is provided in Appendix A in S1 File.

Although binary coding may simplify the complex nature of human error mechanisms, the level of detail available in official investigation reports does not allow reliable quantitative measurement of the relative severity or contribution of individual factors. Given this limitation, an all-or-nothing coding scheme was considered the most appropriate and objective approach for identifying the presence of contributory human and organizational factors across cases. This strategy is consistent with prior accident analysis studies that rely on investigation reports as primary data sources and aim to capture factor occurrence rather than magnitude.

The overall coding workflow is illustrated in Fig 3, which depicts the sequential procedure by which qualitative descriptions are reviewed, highlighted, assigned to predefined categories, and subsequently encoded into binary variables. In addition to showing the conversion process, Fig 3 also illustrates how coded variables are compiled into a structured database, enabling consistent storage, retrieval, and comparative analysis of accident cases. By representing each human error factor as a standardized indicator grounded in both theoretical frameworks (Table 1) and report-based evidence (Table 2), the resulting data set supports transparent and reproducible analysis and provides a consistent basis for applying regression-based methods to evaluate the association between human error factors and accident severity.

### 3.4 Data description

The data set used in this study consists of 105 train accident cases compiled from official investigation reports. Each case is treated as a single observation, and the primary dependent variable is accident cost, expressed in its original monetary value prior to any transformation. Accident costs exhibit substantial variation across cases, capturing the wide range of financial consequences associated with train accidents.

**Table 2. Operational definitions and report-based evidence for coding categories (C1–C8).**

| Code | Description | Representative Investigation Findings |
|---|---|---|
| C1 | Deficiencies in Supervisory and Managerial Control of Safety-Critical Activities | The station operations manager handled train entry without complying with operating regulations, resulting in derailment |
| C2 | Inadequate Organizational Training and Competency Development | Insufficient training on operational rules and driving regulations |
| C3 | Deficiencies in Maintenance, Inspection, or Technical System Integrity | Track gauge and cross-level measurements showed that the cross-level in a curved section exceeded the allowable limit |
| C4 | Inadequacies in Equipment, Emergency Systems, or Human-System Interfaces | The transition curve was constructed shorter (20m) than the design standard (30m) due to topographical constraints |
| C5 | Noncompliance With or Circumvention of Safety Regulations and Procedures | The train was operated with a load exceeding the allowable limit |
| C6 | Failures in Communication, Coordination, or Information Transfer | The dispatcher provided incorrect speed (45 km/h) information to the rescue train (25 km/h) |
| C7 | Deficient Operational Decision-Making or Task-Related Competence | After the initial accident, the local controller failed to display a stop signal |
| C8 | Performance Degradation Due to Fatigue or Physical and Health Conditions | The train driver ignored the stop signal due to drowsiness and reduced alertness |

**Table 3. Inter-rater agreement for human error categories (Cohen's kappa).**

| Category | C1 | C2 | C3 | C4 | C5 | C6 | C7 | C8 | Mean | Range |
|---|---|---|---|---|---|---|---|---|---|---|
| Cohen`s κ | 0.981 | 0.621 | 0.943 | 0.961 | 0.935 | 0.918 | 0.854 | 0.852 | 0.883 | 0.621-0.981 |

**Note.** $\kappa \geq 0.80$ indicates almost perfect agreement, and $0.60 \leq \kappa < 0.80$ indicates substantial agreement.

Table 4 presents the descriptive statistics of the dependent variable together with the eight independent indicators of human error. The results indicate that supervisory and management-related deficiencies are the most frequently observed factor, while health-related conditions appear only rarely. Overall, the data set captures both high-cost accidents with significant human error involvement and lower-cost events with fewer contributory factors, thereby providing a comprehensive empirical basis for subsequent statistical analysis.

## 4. Methodology

### 4.1 Log transformation of the dependent variable

The dependent variable of this study is accident cost, initially expressed in its original monetary values. Preliminary inspection reveals that the raw distribution of costs is heavily right-skewed, with a small number of accidents accounting for extremely high losses. As shown in Fig 4, the raw distribution deviates substantially from normality.

This distributional pattern highlights the need for modeling approaches that can accommodate strictly positive and highly skewed outcomes. Accordingly, in the main analysis, this study adopts a Gamma generalized linear model (GLM) with a log link, which is specifically designed for such data structures.

For comparison and robustness checks, a log-linear specification is also considered. To this end, the natural logarithm of accident cost is applied. This transformation compresses the scale of extreme values and results in a distribution that more closely approximates normality, as illustrated in the right panel of Fig 4.

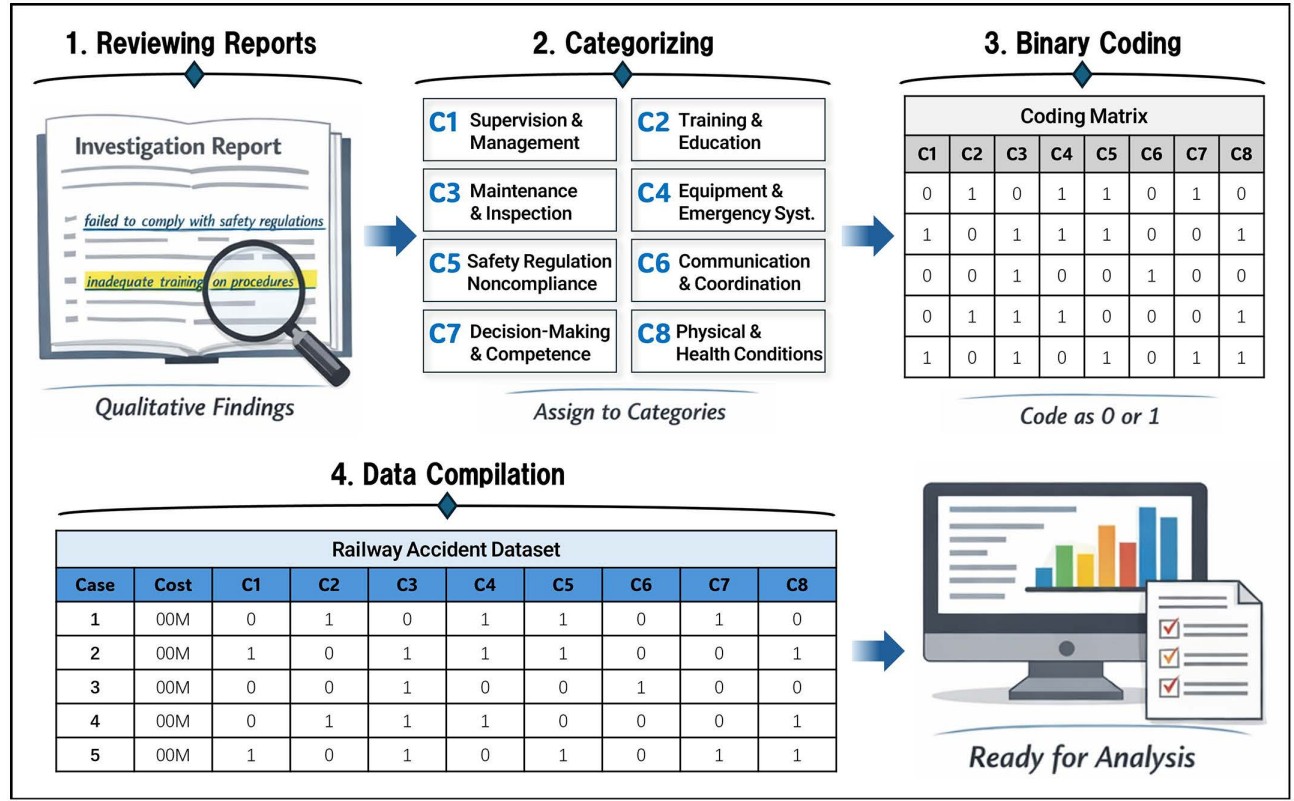

**Fig 3. Workflow for extracting, categorizing, and coding accident report data.**

**Table 4. Descriptive statistics of variables (N = 105).**

| Variable | Description | Mean (Prevalence) | Std. Dev. | Min – Max |
|---|---|---|---|---|
| Cost | Accident cost (KRW, original value) | 144.46M | 250.62M | 0.17M – 1593.00M |
| C1 | Deficiencies in Supervisory and Managerial Control of Safety-Critical Activities (0/1) | 0.43 | 0.50 | 0 - 1 |
| C2 | Inadequate Organizational Training and Competency Development (0/1) | 0.08 | 0.27 | 0 - 1 |
| C3 | Deficiencies in Maintenance, Inspection, or Technical System Integrity (0/1) | 0.48 | 0.50 | 0 - 1 |
| C4 | Inadequacies in Equipment, Emergency Systems, or Human-System Interfaces (0/1) | 0.42 | 0.50 | 0 - 1 |
| C5 | Noncompliance With or Circumvention of Safety Regulations and Procedures (0/1) | 0.32 | 0.47 | 0 - 1 |
| C6 | Failures in Communication, Coordination, or Information Transfer (0/1) | 0.12 | 0.33 | 0 - 1 |
| C7 | Deficient Operational Decision-Making or Task-Related Competence (0/1) | 0.24 | 0.43 | 0 - 1 |
| C8 | Performance Degradation Due to Fatigue or Physical and Health Conditions (0/1) | 0.03 | 0.17 | 0 - 1 |

The improvement is further supported by the Q–Q plots presented in Fig 5, which compare the distributions before and after transformation. The log-transformed variable aligns more closely with the theoretical normal line, with only minor deviations in the tails. These results indicate that the log transformation provides a reasonable approximation for alternative log-linear specifications, which are used as supplementary analyses alongside the baseline Gamma GLM.

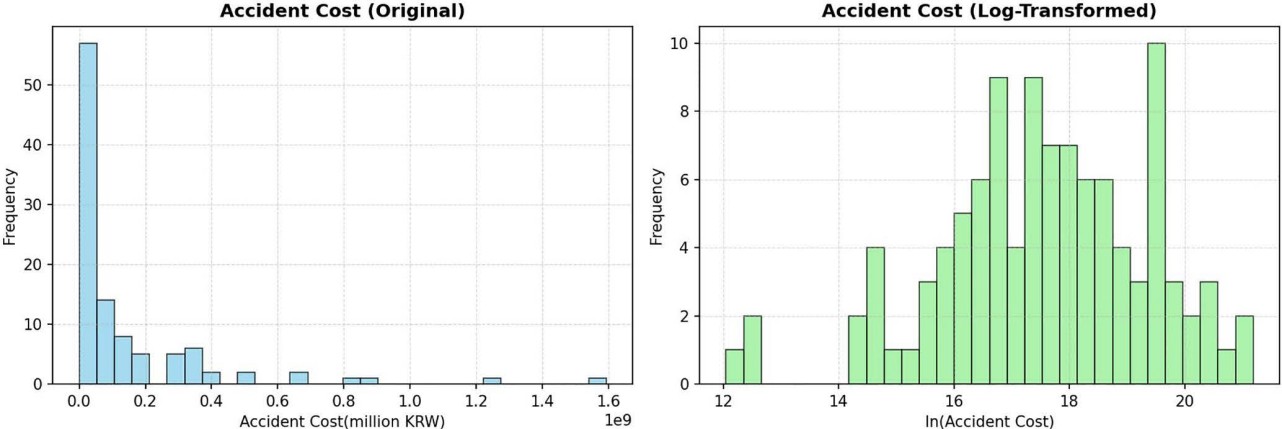

**Fig 4. Distribution of accident cost before and after log transformation.**

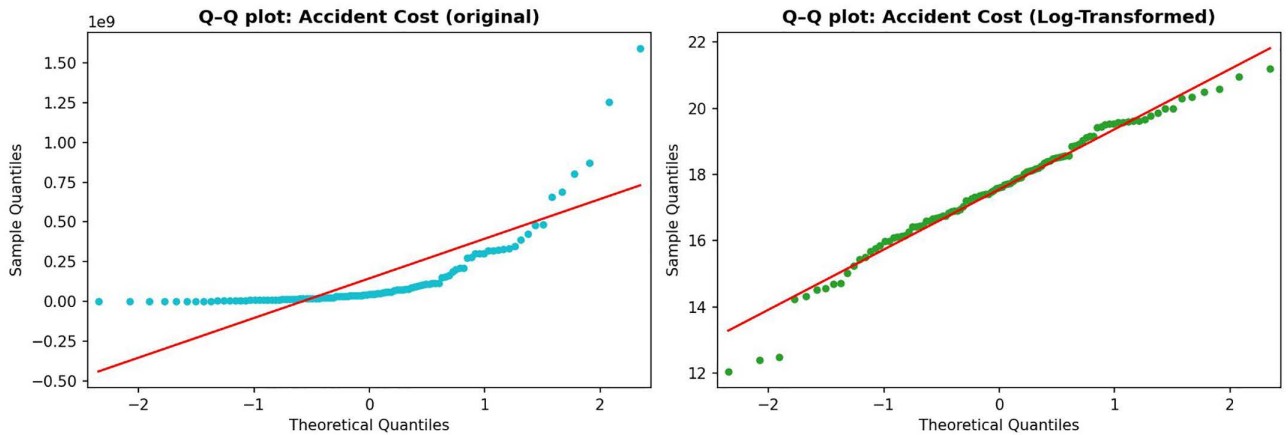

**Fig 5. Q–Q plots of accident cost before and after log transformation.**

## 4.2 Multicollinearity and co-occurrence diagnostics

Before conducting the regression analyses, potential dependence among the explanatory variables was carefully examined to assess the risk of multicollinearity and related identification problems. As a first step, we evaluated linear multicollinearity using the Variance Inflation Factor (VIF) [40]. VIF values indicate how much the variance of an estimated coefficient is inflated due to linear dependence among regressors. Common rules of thumb suggest that values exceeding 10 indicate severe multicollinearity, while more conservative thresholds such as 5 are often applied in safety-related research [41].

The VIF results for the eight human error variables (C1–C8) all fall well below the conservative threshold of 5, indicating that no serious linear multicollinearity is present. This confirms the suitability of including all variables simultaneously in the regression models without the need for dimensionality reduction or variable aggregation. The detailed VIF results are reported in Table 5.

Because the explanatory variables are binary indicators, linear collinearity diagnostics alone may not fully capture their joint occurrence structure. We therefore additionally examined pairwise co-occurrence patterns among C1–C8 using

**Table 5. VIF results for human error variables (C1–C8).**

| Variable | VIF | Tolerance | Variable | VIF | Tolerance |
|---|---|---|---|---|---|
| C1 | 1.06 | 0.94 | C5 | 1.31 | 0.77 |
| C2 | 1.27 | 0.79 | C6 | 1.27 | 0.79 |
| C3 | 1.10 | 0.90 | C7 | 1.31 | 0.76 |
| C4 | 1.31 | 0.90 | C8 | 1.10 | 0.91 |

phi correlation coefficients. Fig 6 presents both a heat-map and the corresponding correlation matrix for the pairwise phi correlations among the eight human error categories. The figure shows that all off-diagonal associations are modest in magnitude, with all absolute values remaining below 0.35, and no block structure or clustering pattern is observed.

Taken together, these diagnostics suggest that the human error categories are not strongly dependent on one another, either in a linear sense or in terms of joint occurrence patterns. This diagnostic evidence indicates that any lack of statistical significance observed in subsequent regression models is unlikely to be primarily attributable to masking effects due to multicollinearity or systematic co-occurrence among the explanatory variables.

### 4.3 Model performance comparison

To determine an appropriate modeling framework for accident cost severity, this study adopts a two-step specification strategy that explicitly accounts for both the distributional characteristics of the dependent variable and the robustness of the estimation results.

First, to assess the appropriate distributional form for accident cost, a Gamma generalized linear model (GLM) with a log link is compared with a log-linear OLS model. The Gamma GLM is estimated on accident cost in levels under a Gamma likelihood, whereas the log-linear OLS model is estimated on ln(COST) assuming normally distributed errors. Because these two models rely on different likelihood functions and are defined on different scales of the dependent

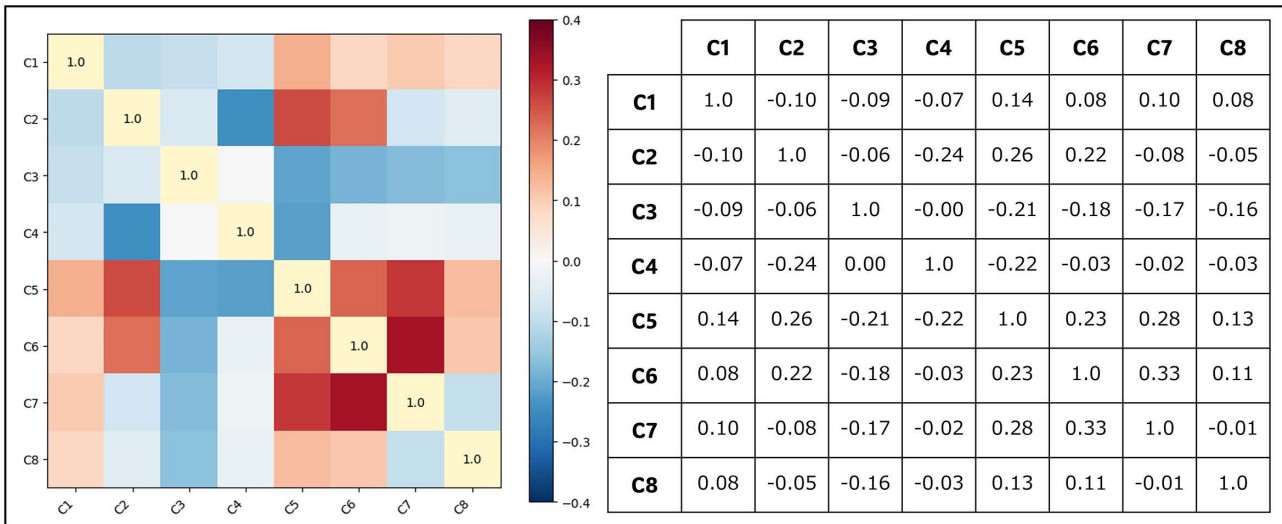

**Fig 6. Pairwise phi correlations among C1–C8 shown as a heatmap and a correlation matrix.**

variable, their goodness-of-fit statistics are not strictly comparable in a formal sense, and any direct comparison of likelihood-based criteria should therefore be interpreted with caution.

Nevertheless, likelihood-based measures such as the log-likelihood, AIC, and deviance are reported to verify that the Gamma specification fits the data in a numerically stable and well-behaved manner (Table 6). The results show that the Gamma GLM converges properly and yields finite and stable fit statistics, indicating that the model does not suffer from numerical instability or obvious misspecification.

To complement this assessment, we additionally examine predictive performance on the original cost scale using the root mean squared error (RMSE) and the mean absolute error (MAE). Although this comparison is not fully symmetric due to the retransformation required for the log-linear OLS predictions, these measures provide a practical benchmark for model performance. As reported in Table 6, the Gamma GLM achieves a lower RMSE (2.34e8) than the log-linear OLS model (2.61e8), whereas the log-linear OLS model yields a lower MAE (1.27e8) than the Gamma GLM (1.46e8). Thus, the two models exhibit broadly comparable predictive performance, with no decisive dominance of either specification in terms of overall prediction error.

Taken together, these results indicate that the choice of the Gamma GLM is not driven primarily by superior predictive accuracy, but rather by its theoretical and empirical suitability for modeling a positive, right-skewed, and highly heterogeneous cost variable. Given the strong skewness and heteroskedastic nature of accident cost data, the Gamma GLM with a log link provides a more appropriate distributional framework for the dependent variable and is therefore adopted as the baseline specification in this study.

Second, within the log-linear framework, OLS, Ridge, and Lasso regressions are compared as robustness checks. All three models use ln(COST) as the dependent variable, but Ridge and Lasso impose L2 and L1 regularization, respectively, with the regularization parameter selected by cross-validation. Because regularized estimators do not have a standard definition of R² and are not estimated by maximum likelihood in a directly comparable way, model performance is evaluated using in-sample and cross-validated prediction errors on the log scale, such as RMSE and MAE (Table 7).

As reported in Table 7, the three models exhibit very similar predictive performance. OLS achieves the lowest in-sample RMSE (1.673) and MAE (1.362), while Ridge yields a slightly lower cross-validated RMSE (1.790) than OLS (1.841) and Lasso (1.801). However, the differences across models are small, indicating that the results are not sensitive to the choice of regularization and are not driven by overfitting or multicollinearity within the log-linear framework.

Taken together, these results support the use of the Gamma GLM with a log link as the baseline specification. This choice is primarily motivated by the strictly positive and highly right-skewed nature of accident cost and is further reinforced by the satisfactory and stable goodness-of-fit of the Gamma specification. The log-linear models are retained as supplementary specifications for robustness checks.

**Table 6. Comparison of distributional specifications for accident cost.**

| Model | DV | Dist. | LogLik | AIC | BIC | RMSE | MAE |
|---|---|---|---|---|---|---|---|
| **Gamma GLM** | Cost | Gamma | −2045.43 | 4110.86 | −229.30 | **2.34e8** | 1.46e8 |
| **OLS_Log (*In* Cost)** | *In* Cost | Normal | −203.09 | 424.17 | 448.06 | 2.61e8 | 1.27e8 |

**Table 7. Robustness checks using regularized log-linear models.**

| Model | DV | RMSE(log scale) | MAE (log scale) | CV RMSE (log scale) |
|---|---|---|---|---|
| **OLS** | *In* Cost | **1.673** | **1.362** | 1.841 |
| **Ridge** | *In* Cost | 1.729 | 1.373 | 1.790 |
| **Lasso** | *In* Cost | 1.818 | 1.426 | 1.801 |

The baseline Gamma GLM specification can be expressed as follows:

$$E(Cost_i) = exp(\beta_0 + \beta_1 X_{1_i} + \beta_2 X_{2_i} + \cdots + \beta_k X_{k_i})$$

(1)

Notes on the model specification:

$Cost_i$: accident cost for observation $i$ (dependent variable).

$X_1, \ldots, X_k$: explanatory variables included in the model.

$\beta_j$: estimated regression coefficients.

This formulation ensures strictly positive fitted values and allows the variance of accident cost to increase with its mean, which is consistent with the empirical properties of the data. It therefore provides a theoretically appropriate and empirically robust framework for examining the factors associated with accident cost severity.

## 5. Results

Following the comparative evaluation of alternative specifications, the Gamma generalized linear model (GLM) with a log link was adopted as the baseline model for accident cost analysis. This choice is motivated by the strictly positive and highly right-skewed nature of accident cost and is supported by the stable goodness-of-fit of the Gamma specification. Log-linear OLS and its regularized variants are retained as supplementary models for robustness checks.

The estimation results of the baseline Gamma GLM are reported in Table 8. The dependent variable is accident cost in levels, and eight explanatory variables representing human error categories (C1–C8) are included in the model. The table reports coefficient estimates, standard errors, z-statistics, and p-values. Because a log link is used, the coefficients can be interpreted in multiplicative terms: exp(β) represents the proportional change in expected accident cost associated with a one-unit change in the explanatory variable.

Among the explanatory variables, C1 (Supervisory and Managerial Control) and C6 (Communication and Coordination Failures) exhibit positive and statistically significant coefficients, indicating that accidents involving organizational and coordination failures are associated with substantially higher accident costs. In particular, the estimated coefficients imply that the presence of C1 is associated with roughly a twofold increase in the expected accident cost, while the presence of C6 is associated with an increase of approximately 2.5 to 3 times. By contrast, C8 (Fatigue and Physical/Health Conditions) shows a statistically significant negative coefficient.

Importantly, the negative coefficient on C8 should not be interpreted as implying that physical or health-related problems reduce accident risk. Rather, conditional on other technical and organizational factors, accidents primarily attributed to such individual-level conditions tend to involve smaller material and operational losses. In contrast, accidents driven by

**Table 8. Baseline regression results (Gamma GLM).**

| Variable | Coefficient | Std. Error | z-Statistic | p-Value |
|---|---|---|---|---|
| Constant | 17.851 | 0.313 | 57.117 | 0.000 |
| **C1 (Supervisory and Managerial Control)** | 0.898 | 0.282 | 3.185 | 0.001*** |
| C2 (Training and Competency Development) | 0.674 | 0.575 | 1.171 | 0.242 |
| C3 (Maintenance and Inspection) | 0.529 | 0.285 | 1.854 | 0.064 |
| C4 (Equipment and Emergency Systems) | 0.295 | 0.289 | 1.021 | 0.307 |
| C5 (Safety Regulation Noncompliance) | −0.292 | 0.331 | −0.882 | 0.378 |
| **C6 (Communication Failures)** | 0.979 | 0.462 | 2.118 | 0.034* |
| C7 (Decision-Making and Task Competence) | −0.161 | 0.364 | −0.441 | 0.659 |
| **C8 (Fatigue and Health Conditions)** | −2.347 | 0.854 | −2.749 | 0.006** |

**Note: Statistical significance is evaluated at the 0.05 (*), 0.01 (**) and 0.001(***) levels.**

organizational, technical, and systemic failures are much more likely to escalate into large-scale, high-cost events. This interpretation is also consistent with the fact that the cost measure used in this study does not monetize human injuries and therefore primarily reflects material and operational damage.

Other factors, including C2 (Training and Competency Development), C3 (Maintenance and Inspection), C4 (Equipment and Emergency Systems), C5 (Safety Regulation Noncompliance), and C7 (Decision-Making and Task Competence), do not reach conventional levels of statistical significance in the Gamma GLM, suggesting that their effects on accident cost are either weaker or more context-dependent once organizational and systemic factors are taken into account.

Robustness checks based on the log-linear OLS, Ridge, and Lasso specifications yield qualitatively similar patterns in terms of coefficient signs and relative importance, confirming that the main conclusions do not depend on the specific estimation method.

Overall, these results provide consistent evidence that accident cost severity is more strongly associated with organizational and systemic deficiencies rather than with individual-level conditions alone, reinforcing the view that high-cost accidents in complex railway systems are rooted in higher-level structural and organizational weaknesses.

## 6. Discussion and conclusion

This study examined the role of human error factors in shaping the severity of train accidents using 105 official investigation reports. Using a structured coding scheme as an analytical tool to transform qualitative investigation findings into quantitative variables, the analysis provides consistent evidence that organizational and technical deficiencies—particularly supervisory failures and coordination problems—are most strongly associated with higher accident costs. These results reinforce the view that accident severity is not merely a consequence of front-line worker mistakes, but is deeply rooted in higher-level systemic and organizational structures.

An especially important insight from the empirical results is the contrast between organizational/systemic factors and individual-level conditions. While organizational and coordination failures are associated with substantially higher accident costs, accidents primarily attributed to individual physical or health conditions tend to involve significantly smaller material and operational losses. This pattern should not be interpreted as implying that individual-level problems are unimportant for safety. Rather, it suggests that accidents rooted in organizational and systemic failures are far more likely to escalate into large-scale and high-cost events, whereas individual-level failures more often correspond to relatively localized and less costly incidents. This interpretation is also consistent with the fact that the cost measure used in this study does not monetize human injuries and therefore primarily reflects material and operational damage.

The findings carry both theoretical and practical implications. Theoretically, the study demonstrates how qualitative accident reports can be systematically converted into structured variables, not for the purpose of validating the classification scheme itself, but to enable comparative and reproducible empirical analysis of different types of human and organizational errors within sociotechnical systems. By bridging narrative data with statistical modeling, this research enhances methodological rigor and expands the scope of railway safety studies beyond descriptive or anecdotal approaches. Practically, the results highlight that effective safety strategies must extend beyond front-line worker performance to encompass organizational oversight, coordination mechanisms, preventive maintenance, and infrastructure reliability. For regulators and railway operators, this implies that investments in supervisory systems, organizational monitoring, and preventive maintenance regimes are likely to yield higher returns in terms of accident loss reduction than measures aimed solely at training or penalizing individual workers.

A key methodological contribution of this study lies in its integration of narrative-based accident investigation reports with quantitative modeling techniques. This approach not only strengthens analytical robustness but also illustrates a transferable analytical procedure that can be applied to other high-risk sectors such as aviation, maritime transport, and

nuclear energy. By demonstrating how qualitative accident narratives can be converted into structured data for empirical testing, the study opens up new opportunities for comparative cross-industry analyses and broader applications of human error quantification.

These results should be interpreted in light of several limitations inherent in the institutional and data environment of railway accident investigation. The data set is limited to officially investigated accidents, reflecting the institutional context in which serious accidents are legally required to be investigated and disclosed, whereas minor incidents are treated as internal safety materials and are not publicly accessible. As a result, the data set provides near-comprehensive coverage of severe accidents, but does not represent the full universe of all railway incidents. This implies a selection effect toward high-severity events, and the estimated relationships should therefore be interpreted as associations conditional on serious accidents, rather than as population-average effects across the entire spectrum of incidents.

With respect to accident costs, although formal guidelines for cost estimation exist, the reconstruction of costs from investigation reports inevitably involves a degree of investigator judgment, which introduces measurement noise. From a statistical perspective, this primarily affects the precision and stability of coefficient estimates rather than their qualitative direction, implying that the reported magnitudes should be interpreted as approximate associations rather than exact causal effect sizes. As the number of observations increases in future studies, such idiosyncratic variation is expected to be partially averaged out.

Finally, the binary coding of human and organizational factors reflects the structure of the current accident investigation system itself, which records causal factors mainly in terms of their presence or absence rather than their intensity or degree. This institutional constraint makes it difficult to model continuous gradients of risk and implies that the estimated effects capture the impact of the existence of a given factor, not the marginal effect of its severity. Consequently, the analysis likely understates the underlying complexity and heterogeneity of human and organizational error mechanisms.

In conclusion, within the scope of officially investigated serious accidents, organizational and technical deficiencies remain the most critical factors associated with accident severity despite advances in railway technology and regulation. By systematically transforming accident reports into structured evidence and applying appropriate statistical models, this study provides a foundation for evidence-based strategies aimed at reducing high-consequence accidents.

The findings highlight the importance of shifting the focus of railway safety management from reactive, individual-centered interventions toward proactive, systemic improvements. Taken together, the results should be read as providing robust evidence about the direction and relative importance of systemic factors, rather than as precise estimates of their population-wide causal magnitudes or as universally generalizable effects across all types of railway incidents. Ultimately, addressing organizational and technical weaknesses is likely to deliver more sustainable safety outcomes and to strengthen the long-term resilience of railway transport systems.

## Supporting information

**S1 File. Appendix.**
(DOCX)

## Author contributions

**Conceptualization:** Changhun Kim.

**Data curation:** Changhun Kim.

**Formal analysis:** Changhun Kim.

**Investigation:** Changhun Kim.

**Methodology:** Changhun Kim.

**Validation:** Jun Lee.

**Visualization:** Changhun Kim.

**Writing – original draft:** Changhun Kim.

**Writing – review & editing:** Changhun Kim, Jun Lee.

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
