## [Decision Letter · Decision Letter 0]

2 Jan 2026

Dear Dr. Lee,

Thank you for submitting your manuscript to PLOS ONE. After careful consideration, we feel that it has merit but does not fully meet PLOS ONE’s publication criteria as it currently stands. Therefore, we invite you to submit a revised version of the manuscript that addresses the points raised during the review process.

We look forward to receiving your revised manuscript.

Kind regards,

Qing-Chang Lu

Academic Editor

PLOS One

Journal Requirements:

3. Thank you for uploading your study's underlying data set. Unfortunately, the repository you have noted in your Data Availability statement does not qualify as an acceptable data repository according to PLOS's standards.

At this time, please upload the minimal data set necessary to replicate your study's findings to a stable, public repository (such as figshare or Dryad) and provide us with the relevant URLs, DOIs, or accession numbers that may be used to access these data. For a list of recommended repositories and additional information on PLOS standards for data deposition, please see https://journals.plos.org/plosone/s/recommended-repositories ..

This research was supported by a Korean Agency for Infrastructure Technology Advancement (KAIA) grant funded by the Ministry of Land, Infrastructure and Transport (grant no. RS-2023-00239464).

Reviewers' comments:

Reviewer's Responses to Questions

**Comments to the Author**

1. Is the manuscript technically sound, and do the data support the conclusions?

Reviewer #1: Yes

Reviewer #2: Partly

Reviewer #3: Yes

2. Has the statistical analysis been performed appropriately and rigorously?

Reviewer #1: Yes

Reviewer #2: No

Reviewer #3: Yes

3. Have the authors made all data underlying the findings in their manuscript fully available?

Reviewer #1: No

Reviewer #2: Yes

Reviewer #3: Yes

4. Is the manuscript presented in an intelligible fashion and written in standard English?

Reviewer #1: Yes

Reviewer #2: Yes

Reviewer #3: No

Reviewer #1: The paper addresses a significant and relevant issue in railway safety: the need for a systematic framework to analyze human error from accident reports. The objective is clear, and the proposed approach has the potential to make a valuable contribution to the field. However, several aspects of the manuscript require clarification and strengthening before it can be considered for publication.

1.While the introduction effectively identifies a historical gap in the literature (citing sources from 2002, 2009, and 2011), it would be more compelling if the authors could demonstrate that this gap persists today. I would recommend including a brief discussion of more recent literature.

2. The methodology section mentions that investigation reports were selected based on “incidents of significant safety and operational relevance.” This is too vague. The authors should provide precise inclusion and exclusion criteria.

3.The paper describes a “structured extraction and coding procedure” but provides no details on its implementation.

4. The reference list requires careful proofreading.

5.The snippets suggest the paper’s contribution is a new coding framework. It is unclear if the paper goes on to apply this framework and present findings from the analysis .

Reviewer #2: The manuscript addresses an important and understudied aspect of railway safety: the contribution of human error factors to the severity of railway accidents. The use of a decade of ARAIB investigation reports and the attempt to systematically code contributory factors represent valuable contributions.

However, several methodological issues require attention before the manuscript can be considered for publication.

Below I provide detailed comments aligned with the assessment questions.

1. Technical Soundness

Your approach - extracting structured variables from narrative accident investigation reports and modeling accident costs using a log-linear OLS model - is conceptually appropriate and of potential interest for safety research. However, the technical soundness of the study is only partially demonstrated.

Model assumptions are not validated.

The manuscript does not include regression diagnostics such as residual plots, normality tests, tests for heteroscedasticity, or influence analysis. Without these, the validity of the OLS model cannot be confirmed.

Model explanatory power is weak.

The adjusted R2 (ca 0.08) indicates that the predictors explain only a small fraction of variance. While this does not invalidate the analysis, it requires explicit discussion.

Potential omitted variable bias.

Important covariates such as accident type, operational environment, temporal trends, rolling stock characteristics, and contextual factors are not included in the model. This omission may bias coefficient estimates and weaken interpretative claims.

Binary coding oversimplifies complex causal factors.

Contributory factors are coded as 0/1, which does not reflect severity or degree of contribution and may flatten the nuances present in the investigation reports.

Given these issues, the conclusions must be framed more cautiously, emphasizing associations rather than causal implications.

2. Statistical Analysis

The statistical strategy-log transformation, VIF analysis, comparison of alternative models-is a good start. However, further rigor is required.

The absence of diagnostic tests prevents evaluation of OLS assumptions.

No robustness checks (e.g., alternative specifications, sensitivity analyses, cross-validation) are presented.

The comparison with Poisson and Negative Binomial models could be clarified, as pseudo-R2 measures are not directly comparable to adjusted R2 in continuous-outcome models.

The coding procedure would benefit from an assessment of inter-coder reliability or clearer coding rules.

Strengthening the statistical validation would significantly improve the paper.

3. Major Conceptual Issue: “Causation” vs “Severity”

The manuscript repeatedly refers to “accident causation” in the title, abstract, and narrative.

However, the dependent variable used in the statistical model is accident cost, which measures severity rather than causation.

Severity and causation are conceptually distinct: a factor can cause an accident without increasing its cost, and vice versa.

I recommend that the authors:

revise the title, abstract, and framing to reflect that the study analyzes severity (cost impacts), not causation;

avoid causal language (e.g., “determinants”, “drivers”);

consider a more accurate title such as:

“Assessing the Impact of Human Error Factors on Railway Accident Severity…”

This adjustment would substantially improve conceptual clarity and align the paper with its actual analytical content.

4. Additional Suggestions for Improvement

Expand the limitations section.

You should explicitly acknowledge:

- the limited generalizability of officially investigated accidents,

- potential biases in investigator-defined contributory factors,

- limitations arising from binary coding,

- the low explanatory power of the model.

Provide concrete examples of coding.

Including short excerpts from investigation reports illustrating how specific statements were mapped to categories C1 - C8 would greatly enhance transparency.

Consider additional covariates in future models.

Even a few basic controls (accident type, operator, year, infrastructure type) would improve model robustness.

Interpret findings with greater caution.

Some statements imply causality ("key determinants", "drivers"), whereas the study design supports associations only. Rephrasing would improve accuracy.

5. Ethical and Editorial Issue: Inconsistent Funding Disclosure

There is an inconsistency in the funding information that should be corrected.

In the submission metadata, the authors state that “the authors received no specific funding for this work”.

However, in the Acknowledgements section of the manuscript, the authors report support from the Korean Agency for Infrastructure Technology Advancement (KAIA), grant RS-2023-00239464.

This discrepancy should be corrected to ensure compliance with PLOS funding disclosure requirements. The funding source should be declared consistently in both the metadata and the manuscript.

6. Overall Evaluation

This is a promising manuscript on an important topic, and the dataset represents a valuable resource. However, methodological and analytical limitations need to be addressed before the results can be considered robust. Strengthening the statistical validation, expanding the limitations, and clarifying aspects of the coding scheme will enhance both the credibility and the contribution of the study.

Reviewer #3: The peer-review report and the language-editing corrections have been provided and submitted in a separate file. Please refer to the attached document for the full set of reviewer comments as well as detailed, line-by-line English language revisions and consistency fixes (spelling, punctuation, and terminology).

**Do you want your identity to be public for this peer review?** For information about this choice, including consent withdrawal, please see our For information about this choice, including consent withdrawal, please see our Privacy Policy .

Reviewer #1: No

Reviewer #2: **Yes:** Lorenzo FedeleLorenzo Fedele

Reviewer #3: No

---

## [Author Response · Author response to Decision Letter 1]

28 Jan 2026

We have carefully revised the manuscript in accordance with all comments and requests from the Academic Editor and the reviewers, and we believe that all issues raised in the decision letter have now been fully addressed. Please refer to the accompanying “Response to Reviewers” document for a detailed, point-by-point explanation of how each comment was handled.

---

## [Decision Letter · Decision Letter 1]

10 Mar 2026

Assessing the impact of human error factors on railway accident severity: Evidence from accident investigation reports in Korea

PONE-D-25-51103R1

Dear Dr. Lee,

We’re pleased to inform you that your manuscript has been judged scientifically suitable for publication and will be formally accepted for publication once it meets all outstanding technical requirements.

Kind regards,

Qing-Chang Lu

Academic Editor

PLOS One

Additional Editor Comments (optional):

Reviewers' comments:

Reviewer's Responses to Questions

**Comments to the Author**

Reviewer #1: All comments have been addressed

Reviewer #2: All comments have been addressed

2. Is the manuscript technically sound, and do the data support the conclusions?

Reviewer #1: Yes

Reviewer #2: Yes

3. Has the statistical analysis been performed appropriately and rigorously?

Reviewer #1: Yes

Reviewer #2: Yes

4. Have the authors made all data underlying the findings in their manuscript fully available?

Reviewer #1: Yes

Reviewer #2: Yes

5. Is the manuscript presented in an intelligible fashion and written in standard English?

Reviewer #1: Yes

Reviewer #2: Yes

Reviewer #1: The authors have well addressed all my comments, I appreciate the authors' efforts to improve the manuscript.

Reviewer #2: The authors have adequately addressed all the.

The manuscript has substantially improved in terms of clarity, methodological rigor, statistical analysis, and transparency.

I therefore consider the manuscript suitable for publication in its current form.

**Do you want your identity to be public for this peer review?** For information about this choice, including consent withdrawal, please see our For information about this choice, including consent withdrawal, please see our Privacy Policy .

Reviewer #1: **Yes:** Gen LiGen Li

Reviewer #2: **Yes:** Lorenzo FedeleLorenzo Fedele

---

## [Editor Report · Acceptance letter]

PONE-D-25-51103R1

PLOS One

Dear Dr. Lee,

I'm pleased to inform you that your manuscript has been deemed suitable for publication in PLOS One. Congratulations! Your manuscript is now being handed over to our production team.

Kind regards,

on behalf of

Dr. Qing-Chang Lu

Academic Editor

PLOS One